# Cyanide Biodegradation by *Trichoderma harzianum* and Cyanide Hydratase Network Analysis

**DOI:** 10.3390/molecules27103336

**Published:** 2022-05-23

**Authors:** Narges Malmir, Mohammadreza Zamani, Mostafa Motallebi, Najaf Allahyari Fard, Lukhanyo Mekuto

**Affiliations:** 1National Institute of Genetic Engineering and Biotechnology (NIGEB), Shahrak-e Pajoohesh km 15, Tehran-Karaj Highway, Tehran P.O. Box 14965/161, Iran; nargesmalmir1@gmail.com (N.M.); zamani@nigeb.ac.ir (M.Z.); motalebi@nigeb.ac.ir (M.M.); allahyar@nigeb.ac.ir (N.A.F.); 2Department of Chemical Engineering, University of Johannesburg, Johannesburg 2028, South Africa

**Keywords:** cyanide, biodegradation, *Trichoderma harzianum*, cyanide hydratase (CHT)

## Abstract

Cyanide is a poisonous and dangerous chemical that binds to metals in metalloenzymes, especially cytochrome C oxidase and, thus, interferes with their functionalities. Different pathways and enzymes are involved during cyanide biodegradation, and cyanide hydratase is one of the enzymes that is involved in such a process. In this study, cyanide resistance and cyanide degradation were studied using 24 fungal strains in order to find the strain with the best capacity for cyanide bioremediation. To confirm the capacity of the tested strains, cyano-bioremediation and the presence of the gene that is responsible for the cyanide detoxification was assessed. From the tested organisms, *Trichoderma harzianum* (*T. harzianum*) had a significant capability to resist and degrade cyanide at a 15 mM concentration, where it achieved an efficiency of 75% in 7 days. The gene network analysis of enzymes that are involved in cyanide degradation revealed the involvement of cyanide hydratase, dipeptidase, carbon–nitrogen hydrolase-like protein, and ATP adenylyltransferase. This study revealed that *T. harzianum* was more efficient in degrading cyanide than the other tested fungal organisms, and molecular analysis confirmed the experimental observations.

## 1. Introduction

The accumulation of cyanide-containing wastewaters in the environment is a serious hazard for ecosystems and human health due to the toxicity of cyanide. Thus, before the discharge of these effluents into the environment, they need to be remediated, either by physical, chemical, and/or biological methods [1,2,3,4,5]. The general amount of cyanide that is released from industrial activities has been estimated to be approximately 14 million kg/yr. Three forms of cyanide, including hydrogen cyanide and sodium or potassium cyanide (NaCN or KCN, solid), are the main man-made cyanide compounds [6]. Cyanide compounds are toxic and hazardous for most living organisms because of their strong binding to metalloproteins, which ultimately results in the inactivation of the electron transport chain [7,8]. In fact, cyanide can form complexes with toxic metals, such as nickel, copper, zinc, and iron. The stability and resistance to the biodegradation of such cyano-metal complexes are comparable with free cyanide [9]. Natural and anthropogenic activities are a result of cyanide production and presence in the environment, and these compounds can be in different forms, i.e., liquid, solid, and gas phases [6,10]. Organisms such as plants, bacteria, and fungi are some of the natural cyanide producers; however, the main source of cyanides in the environment are from anthropogenic sources, such as in metal extraction, electroplating, polymer synthesis, steel manufacturing, carbonization, organic chemical production, pharmaceutical, and agricultural and gold mining industries [2,11]. These industries generate free- and metal-cyanide complexes at different concentrations and volumes, thus affecting the living organisms that thrive in aquatic environments when cyanide-contaminated waters are discharged into surface waters [12]. 

Cyanide can be removed from industrial wastewater by biodegradation, using physical and chemical methods. Alkaline chlorination, barren water rinse, ozonation, adsorption through granulated activated carbon, and ion exchange are some examples of the physical and chemical methods [13]. Physicochemical degradation methods for the removal of cyanide compounds are expensive and generate additional poisonous products. Moreover, these techniques cannot completely treat some of the cyanide complexes [14]; however, cyanide biodegradation provides an appropriate alternative [15]. The biological degradation process has been studied extensively since it is economical and environmentally friendly [1,16]. Some aerobic micro-organisms are able to utilize thiocyanate (SCN^−^) and cyanideas a nitrogen source or as a sulfur source, in the case of SCN^−^ [17]. Several species of bacteria, fungi, and algae have been reported to be cyanide degraders. Some microbial strains, such as *Klebsiella oxytoca*, *Pseudomonas fluorescens*, *Escherichia coli*, *Fusarium solani*, *Stemphylium loti*, *Rhizopous oryzae*, etc., are capable of useing these compounds as a source of nitrogen and carbon for their own growth [18]. Five common pathways (Hydrolytic, Oxidative, Reductive, Substitution/transfer, Syntheses) are involved in the biodegradation of cyanide and different organisms are able to use one or more pathways for cyanide biodegradation [13]. 

However, major focus has been directed to the hydrolytic pathway for cyanide biodegradation, and this is due to: (a) the specific activity of hydrolytic pathway enzymes is high, (b) the enzymes do not need any cofactors, and (c) the triple nonpeptide bonds in cyanide compounds are cleaved directly and the products have low toxicity and can be degraded further [19]. Cyanide hydratase (CHT), nitrile hydratase (NH), thiocyanate hydrolase (TCH, carbonyl pathways), nitrilase, and cyanidase are the enzymes that are involved in hydrolytic pathways. Some phytopathogenic fungi, such as *S. loti*; *Leptosphaeria maculans*; *Gloeocercospora sorghi*; the *Fusarium* species; and saprotrophic fungi, such as *Neurospora crassa*, *Aspergillus nidulans*, and bacterial species, have an intracellular and inducible enzyme that is known as cyanide hydratase (formamide hydrolyase E.C. 4.2.1.66), which catalyzes the hydration of cyanide to produce formamide [11,20]. Rhodanese (EC 2.8.1.1) and cyanide hydratase (EC.4.2.1.66) are the cyanide-degrading enzymes that are produced by *Trichoderma* sp. [21]. The biodegradation of metal-cyanide compound by Fusarium oxysporum N-10 was aided by cyanide hydratase and the enzyme appears to exist as a homotetramer [22]. The cyanide hydratase of *Aspergillus niger* K10 was expressed in *E.coli*, and it has been shown that HCN and nitriles can be degraded using this enzyme. Additionally, cyanide hydratase and nitrilase activity were not observed in the truncated enzymes that did not contain 18–34 C-terminal amino acids [20]. 

These studies demonstrate that fungal organisms are capable of degrading cyanide species. However, there are minimal studies that have studied various filamentous fungal organisms’ biodegradative capabilities of cyanide, including one study that focused on the detection and sequence alignment of the cyanide hydratase gene. Hence, this present study focused on the assessment of various filamentous fungi for the biodegradation of cyanide, including the detection of the cyanide hydratase gene. Sequence alignment of the cyanide hydratase gene from the assessed fungal organisms and their gene networks with respect to Trichoderma were also undertaken.

## 2. Results

### 2.1. Screening of Cyanide-Degrading Fungi

Some fungal strains can survive in cyanide-containing environments, and they may possess the basic mechanisms to tolerate and degrade cyanide compounds. In this research, 24 fungal isolates were studied for their resistance to cyanide by measuring their growth in a medium that contained cyanide. These fungi were cultured at different concentrations of KCN for 7 days in triplicates (Table 1). Five resistant species, including two isolates of *Fusarium oxysporum* (FO1 and FO2), *Trichoderma harzianum* (TH1), *Alternaria* sp. (A2), and *Chaetomium globosum*, were selected based on their high tolerance to the highest cyanide concentration of 15 mM. Different fungal species that were not affected by cyanide in their surrounding medium have two different strategies; (i) they can tolerate cyanide and utilize other nutrients in the medium for their survival, and (ii) they can use cyanide as a carbon or nitrogen source. The criteria for choosing these five fungi were based on the increasing growth of these organisms compared to the control as cyanide concentrations increased. Figure 1 shows the proportion of fungal growth in 15 mM cyanide-containing media on fungi dry weight, and cultured without cyanide, and this proportion was from 1.4 for *C. globosum* to 5 for *F. oxysporum* (FO1). In this study, the highest proportion range was seen in *F. oxysporum* (FO1) and *T. harzianum* (TH1). Since *T. harzianum* (TH1) is not a phytopathogen and possesses a biocontrolling role in plants, it was selected for further studies. 

For subsequent studies, *T. harzianum* was cultured in the media of 15, 20, 25, and 30 mM of cyanide and also in media without cyanide as a control for 3, 5, and 7 days. The 7-day cultures had the maximum fungal growth and the increase in fungal growth on the 3-day cultures were minimal (Figure 2). These results suggest that *T. harzianum* in the 7-day culture tolerated the cyanide in the surrounding media and used it as a nitrogen and/or carbon source. 

### 2.2. Measurement of Cyanide Degradation

*T. harzianum* was cultured in media containing 15, 20, 25, and 30 mM of cyanide over a period of 3, 5, and 7 days. *T. harzianum* achieved the highest biodegradation efficiency of 75% at a concentration of 15 mM cyanide in the 7-day culture (Table 2). Cyanide degradation by this strain at a 15 mM cyanide concentration over 3, 5, and 7 day periods was higher compared to other tested concentrations, while there were lower degradations at the higher concentrations of 20, 25, and 30 mM. 

### 2.3. Analyzing the Cyanide Detoxification Pathway and Gene Network of Trichoderma and the Effect of Cyanide on Fungal Growth

Cyanide hydratase has been shown to be the key enzyme in the cyanide degradation pathway in previous studies [23]. This study confirmed the importance of cyanide hydratase in pathway determination using KEGG orthology; K10675 in the *Trichoderma* genus. Different proteins, which are involved in cyanide detoxification pathway, confirmed by KEGG terms, show that the cyanide hydratase gene plays a key role in the cyanide degradation pathway. In addition to cyanide hydrate (EC 4.2.1.66), formamidase (EC 3.5.1.49), L-3-cyanoalanine synthase (EC 4.4.1.9), and nitrilase (EC 3.5.5.1) are also involved in the cyanide degradation pathway and have been studied to determine their role in the gene network of *Trichoderma* genus. In fact, formamidase plays an important role in formamide degradation, which is produced during cyanide degradation by cyanide hydratase, and it can be used as a carbon and nitrogen source. As a result, formamide degradation can result in an increase in the growth of the organism. As observed from the data (see Figure 1 and Figure 2, Table 1), cyanide degradation in *Trichoderma* is coupled with the increasing dry weight and the interaction of these proteins and the cause of the increasing dry weight can be investigated through system biology analysis. The gene network of *Trichoderma* is constructed by merging multidatabases that show cyanide hydratase (G0RNY5_HYPJQ), Dipeptidase (G0R9H3_HYPJQ), Dipeptidase (G0RV68_ HYPJQ) in the directed networks, and the Carbon–nitrogen hydrolase-like protein (G0RW75_HYPJQ) and ATP adenylyltransferase (G0RJ02_HYPJQ) in the undirected networks, which are known as a hub (central) proteins (Figure 3). Cyanide hydratase, Dipeptidase, and carbon–nitrogen hydrolase-like proteins target nitrogen–carbon bonds, and cyanide hydratase and carbon–nitrogen hydrolase-protein can effectively degrade cyanide compounds. According to the network, cyanide hydratase is associated with various proteins that are involved in cyanide degradation, and the beta subunit of assimilatory sulfite reductase (G0RAA0_HYPJQ) is connected with cyanide hydratase through the MET3 protein, which is one of the important proteins in fungal growth (Figure 3). These whole connections can lead to the increasing growth of the organism. 

### 2.4. Cloning of Cyanide Hydratase

Since cyanide hydratase plays a major role in cyanide bioremediation in *Trichoderma* [21], this study confirmed the presence of the cyanide hydratase (cht) gene, which was cloned in the pJET1.2 cloning vector, subsequent to the amplification of the genomic DNA using appropriate primers. The confirmed amplified fragment was cloned into pJET1.2, and after gene sequencing analysis, it was revealed that despite the similarity with the cht gene from other strains of *Trichoderma*, this sequence was unique and there was no 100% identity with other cht sequences (Table 3).

### 2.5. Sequence Alignment for Cyanide Hydratase Protein

The cyanide hydratase gene (cht) was aligned with other cht gene from the *Trichoderma* species, including the *Trichoderma gamsii* strain T6085 (Tgcht) (GenBank: XP_018662295.1), *Trichoderma resiee* RUT C-30 (Trcht) (GenBank: ETR99830.1), the *Trichoderma arundinaceum* strain IBT 40837 (Tacht) (GenBank: RFU81705.1), and the *T. harzianum* strain T6776 (Thtcht) (GenBank: KKO98780.1), and has a 75.7, 79.2, 80.6, 92.9% identity with each strain, respectively (Figure 4). The phylogenic tree of cyanide hydratase in these strains indicates the relationship between them (Figure 5).

## 3. Discussion

In this study, the selected fungi, *T. harzianum*, not only had a high tolerance to cyanide but was also able to use cyanide as a nitrogen since its dry weight increased in the 15 mM cyanide concentration medium in the 7-day culture. This notion is deciphered from the fact that the PDB media contained carbon-based compounds without a nitrogen source. Previous studies have demonstrated that a number of microorganisms, such as *Pseudomonas pseudoalcaligenes*, *F. solani*, and *Trichoderma* sp., are able to convert cyanide to ammonia and bicarbonate and use these products as sources of nitrogen and carbon, respectively. The reaction mechanism of cyanide hydratase in *L. maculans* produces formamide, which can also be utilized as a source of nitrogen [21,24,25]. The maximum cyanide degradation that was observed using indigenous *T. harzianum* was 75% at a concentration of 15 mM in the 7-day culture in comparison to other studies with lower cyanide concentrations (see Table 4). The cyanide degradation of 15 mM cyanide in the 5-day and 3-day cultures was greater than the other tested concentrations. However, in the concentration of 20, 25, and 30 mM cyanide, the cyanide degradation efficiency decreased. The decrease in this concentration can be attributed to substrate inhibition and toxicity. It has been reported that different organisms have different tolerances to cyanide concentrations; however, at a particular concentration, cyanide becomes toxic to the organism and can lead to the suppression of the growth of an organism [4].

Previous studies observed the highest free cyanide biodegradation efficiency of 77% for *F. oxysporum* at 22 °C and pH 11 [26]. The cloning of cyanide hydratase, which is one of the cyanide-degrading enzymes that are present in *Trichoderma* sp., and the confirmation of its presence showed its key role in cyanide degradation. *A. niger* K10 was observed to possess the cyanide hydratase gene, which was expressed in *E.coli,* where the *E.coli* that was expressing the cht gene was able to degrade nitriles, thus demonstrating the role that cyanide hydratase plays in cyanide biodegradation [20]. Besides this microorganism, the cyanide hydratase gene exists in some phytopathogenic fungi, such as *S. loti*, *L. maculans*, *G. sorghi*, and the *Fusarium* genus, and saprophytic fungi, such as *N. crassa* and *A. nidulans*, which are organisms that were tested in this study [20]. The sequence alignment of cyanide hydratase isolated from *T. harzianum* using the BLAST tool [27] revealed a 94.01% identity and 99% coverage with cyanide hydratase of *Trichoderma simmonsii*. Additionally, the multiple sequence alignment of the cyanide hydratase of six species showed a 94.01% to 79.01% identity with our query (sequence) and the catalytic triad, C-K-E, was conserved. Liu et al. (2013) carried out multiple sequence alignments of nitrilase sequences and the catalytic triad, which was similar to the one observed in this study, was revealed [28]. The phylogenic tree indicates the same origin for the *Trichoderma harzianum* strain CBS 229.95 and the indigenous *T. harzianum* at the same time, before the other genes emerged. Gene network analysis demonstrated that different proteins are involved in cyanide bioremediation, and these include cyanide hydratase (EC 4.2.1.66), formamidase (EC 3.5.1.49), L-3-cyanoalanine synthase (EC 4.4.1.9), and nitrilase (EC 3.5.5.1). The analysis also revealed that Cyanide hydratase (G0RNY5_HYPJQ), Dipeptidase (G0R9H3_HYPJQ), and Dipeptidase (G0RV68_ HYPJQ) are found in the directed networks, while Carbon–nitrogen hydrolase-like protein (G0RW75_HYPJQ) and ATP adenylyltransferase (G0RJ02_HYPJQ) are found in the undirected networks; these are known as hub (central) proteins. Since the beta subunit of assimilatory sulfite reductase has a key role in growth regulation [29], it can help in the fungal resistance and growth of *Trichoderma* organisms.

## 4. Materials and Methods

### 4.1. Materials, Fungal Strains and Growth Condition

Materials: All material in this study were prepared from the Merck Company (Tehran, Iran). PCR purification kit, restriction enzymes, and Master Mix for PCR amplification were supplied by the Roche and Amplicon/iNtRON Company. DNA Ladder Mix, high pure agarose, and primers were prepared by Fermentase, Invitrogen, and GenFanAvaran, respectively. pJET1.2 plasmids were prepared by the Fermentase Company. In this study, *E. coli DH5α* was used as a bacterial strain for cloning purposes and was supplied by the Invitrogen Company. 

### 4.2. Fungal Strains and Growth Condition

Twenty-four fungal strains were selected from the collection in the National Institute of Genetic Engineering and Biotechnology (NIGEB), and these organisms are shown in Table 5. These fungal strains were selected based on the consulted literature, and all fungal strains were cultured in potato dextrose broth (PDB) medium with 25 µg/mL sephadex as an inhibitor of bacterial growth (pH 9.5) and incubated in a shaking incubator at 30 °C and 180 rpm shaking speed. These strains were maintained on potato dextrose agar (PDA). The PDB contained 4 g/L potato powder and 20 g/L of dextrose, while the PDA consisted of the same contents but was supplemented with microbiological agar so that the media was able to set in a petri dish.

### 4.3. Fungi Screening by Cyanide

The screening of 24 fungal strains for cyanide tolerance and biodegradability was carried out in four concentrations (2, 5, 10, and 15 mM) of cyanide. All the organisms were cultured in potato dextrose agar in petri dishes for 14 days in at room temperature. After the 14-day period, the plates were flooded with sterile distilled water and the mycelia were scraped from the surface of the agar using a sterile inoculation loop. The mycelia were separated from the liquid fraction (contained the spores) by filtration using Whatman No.1 filter paper and were inoculated in cyanide containing potato dextrose broth (PDB) at a concentration of 3 × 10^6^ spores/L. The spore concentration was calculated using a hemacytometer under an optical microscope (Olympus, Pretoria, South Africa). The PDB was supplemented with different concentrations of filter-sterilized KCN (cyanide concentration of 2, 5, 10, and 15 mM). The mixture was incubated at 30 °C, 180 rpm for 7 days. Thereafter, the medium with fungal cultures was filtered through a Whatman No. 1 filter paper and dried at 70 °C for 48 h in order to the measure the fungal dry weight. The selection method for screening was based on fungal growth over time. Thereafter, the candidate fungal organism for cyanide bioremediation was selected based on its nonpathogenicity to plants. The nonpathogenicity of the organisms was determined by consulting the literature. To minimize cyanide volatilization, the falcon tubes were sealed with parafilm. The cyanide-containing media was used for the colorimetric and cyanide degradation assays. The GraphPad Prism 8 was used for drawing the diagrams.

### 4.4. Measurement of Residual Cyanide and Cyanide Degradation by Biochemical Assay Method

Cyanide concentration was quantified calorimetrically using the Picric acid assay [37] at a wavelength of 492 nm, using a standard curve to estimate the cyanide concentration [38]. 

### 4.5. Bioinformatics Analysis

#### Analyzing the Gene Network of Trichoderma and the Effect of Cyanide on Fungal Growth

The categorization of genes involved in cyanide degradation was performed using the KEGG database (https://www.kegg.jp/kegg/genes.html) (accessed on 30 September 2021). For the proteome study and protein analysis, data mining tools from NCBI, and UniProt were used. As the target gene list, 107 nodes were imported into STRING 11 and Cytoscape 3.7 for protein–protein interaction, pathway analysis, network reconstruction, and visualization. Of the pathways, 20 were generated from STRING analysis and the average local clustering coefficient was 0.613. To limit the number of nodes and improve the accuracy of our data, the average local clustering coefficient was adjusted to 0.739 and 27 nodes were imported into STRING and Cytoscape and 11 pathways were generated.

### 4.6. Nucleic Acid Manipulations and Sequence Analysis

#### Cloning of Cyanide Hydratase

DNA was extracted from lyophilized mycelia by an efficient protocol for the isolation of high molecular weight DNA from filamentous fungi [39]. All manipulation of nucleic acids, including proliferation, digestion with restriction endonucleases, cloning, and gel electrophoresis were performed as follows: The proliferation of cyanide hydratase was carried out using Taq DNA polymerase and Pfu DNA polymerase in a polymerase chain reaction (PCR). The primer sequences were designed by Oligo Primer Designer (Table 6) [40]. The PCR product was purified, inserted in a pJET1.2 vector and cloned in *E. coli DH5α*, according to the manufacturer’s and the Sambrook protocol, respectively. The enzymatic digestion process using XhoΙ and NcoΙ was performed to confirm the accuracy of cyanide hydratase gene (cht) cloning. In addition to enzymatic digestion, PCR cloning was carried out successfully. Gene sequencing of cht was performed and published (Genbank Accession Number MH629686).

### 4.7. Sequence Alignment for Cyanide Hydratase Protein

Since the cht gene was characterized from the indigenous *T. harzianum* that was isolated in Kerman, Iran, it has some differences with the other cht genes of Trichoderma genus and its gene and protein sequence was aligned with the sequences that are extant in the Genbank database (with accession number CP075866.1, XM_024916132.1, XM_0140973141.1, XM_006966895.1, XM_024891496.1, XM_018851715.1 for genes and QYS99975.1, XP_024773123.1, XP_013952789.1, XP_006966957.1, XP_024746524.1, XP_018701201.1 for proteins) using ClustalW2 [ClustalW2 program: http://www.ebi.ac.uk/Tools/msa/clustalw2/] (accessed on 30 September 2021) and Jalveiw, and the phylogenic tree was drawn [41]. These sequences and structural alignments were performed in order to identify their similarity and conserved in the cysteine–lysine–glutamate catalytic triad. The method of constructing the phylogenic tree was based on a maximum composite likelihood (MCL) approach using Mega6 software [42].

## 5. Conclusions

The present study aimed to assess the biodegradative capabilities of 24 fungal organisms at different cyanide concentrations through the assessment of fungal growth in each cyanide concentration. It was observed that *T. harzianum* had the highest degradation of 75% at a concentration of 15 mM over a 7-day period. It is also important to emphasize that *F. oxysporum* possessed similar degradation efficiency as *T. harzianum* but was excluded due to its pathogenic traits towards plants. Subsequent to cyanide biodegradation by *T. harzianum*, the presence of the cyanide hydratase gene in *T. harzianum* showed that cyanide degradation was due to the activity of this enzyme. Furthermore, gene network analysis revealed that cyanide hydratase, dipeptidase, carbon–nitrogen hydrolase-like protein, and ATP adenylyl transferase to be the hub proteins, while the beta subunit of the assimilatory sulfite reductase had a positive influence on *T. harzianum* growth. The similarity analysis of cyanide hydratase in different fungi revealed that the catalytic triad is conserved and plays a major role in enzyme activity and degradation. Therefore, *T. harzianum* can be used as a cyanide degrading organism due to its capability for cyanide treatment and using cyanide as a carbon and nitrogen source. Therefore, *T. harzianum* has shown that it is an organism which can be utilized in the treatment of cyanide. However, future research should focus on the application of this organism and the degradation of cyanide in the presence of metals and other contaminants, such as ammonia and nitrates since cyanide is mostly associated with these compounds in wastewaters.

## Figures and Tables

**Figure 1 molecules-27-03336-f001:**
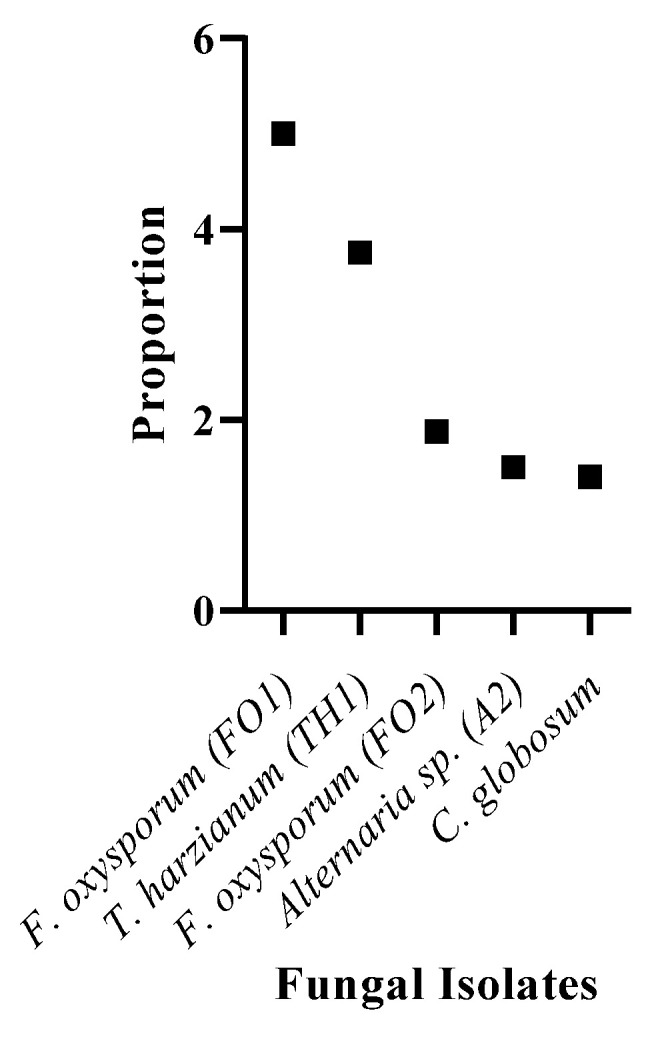
Proportion of fungal dry weight cultured in the media containing 15 mM cyanide.

**Figure 2 molecules-27-03336-f002:**
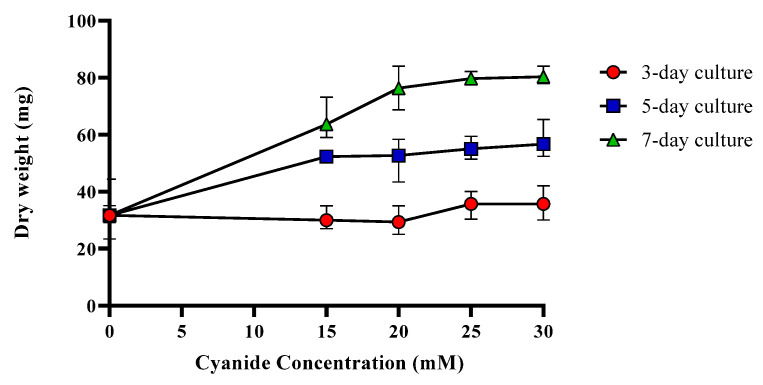
*T. harzianum* dry weight in cyanide-containing media (15, 20, 25, and 30 mM cyanide).

**Figure 3 molecules-27-03336-f003:**
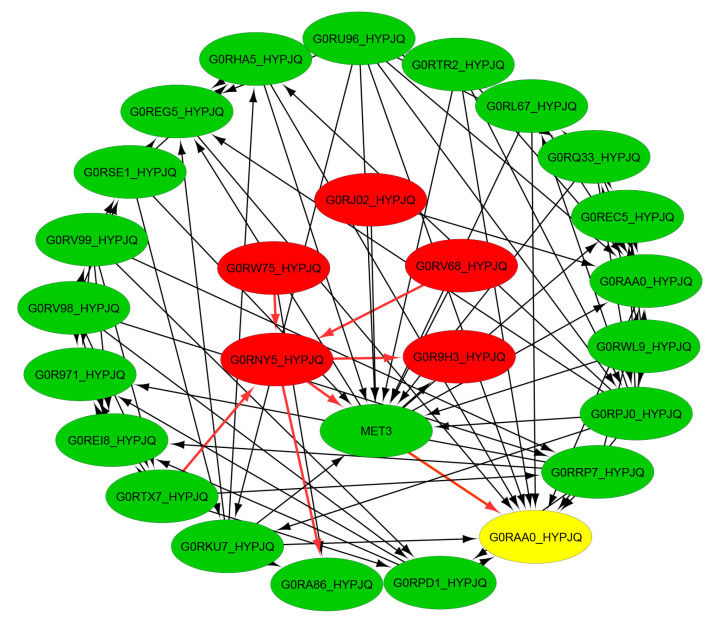
Constructed gene network of cyanide hydratase and some related genes using STRING and Cytoscape 3.7. The red genes symbolize the central network of proteins, while the yellow gene symbolizes the cyanide hydratase encoding gene.

**Figure 4 molecules-27-03336-f004:**
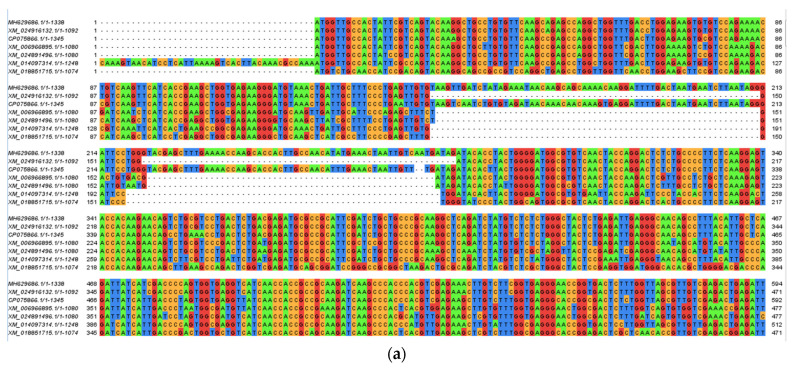
The percentage of identity in the cyanide hydratase of indigenous *T. harzianum* and four Trichoderma species and their catalytic triads, using (**a**) ClustalW and (**b**) Jalview.

**Figure 5 molecules-27-03336-f005:**
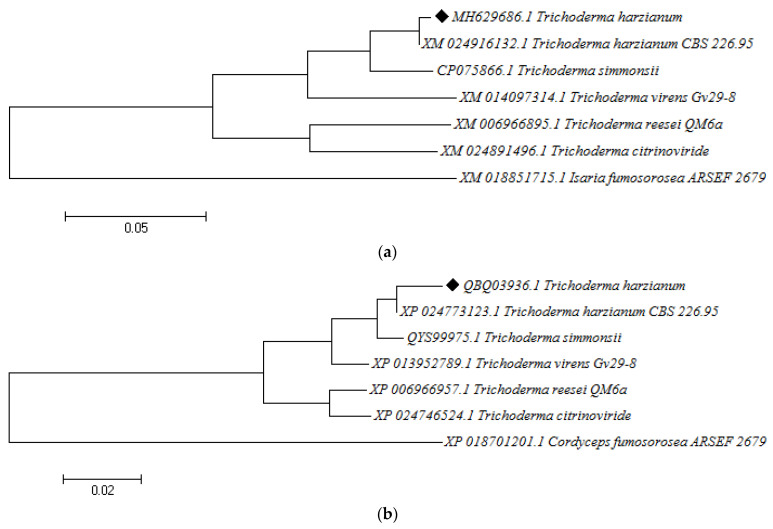
The phylogenic tree of the cyanide hydratase gene (**a**) and protein (**b**) from indigenous *T. harzianum* and four Trichoderma species, using Mega6 software.

**Table 1 molecules-27-03336-t001:** Biomass of 24 fungal isolates cultured in four different cyanide-containing media (2, 5, 10, and 15 mM cyanide), achieved on day 7 of incubation.

No.	Fungus	Average Dry Weight of Fungal Mycelium (mg) Grown in PDB with Different Cyanide Concentrations, Including 0, 2, 5, 10, and 15 mM				
		0 mM	2 mM	5 mM	10 mM	15 mM
1	*Fusarium oxysporum* (FO1)	8	41	39	40	40
2	*Fusarium oxysporum* (FO2)	8	24	29	29.5	32
3	*Fusarium graminearum* (FG1)	19	25	21	19	19
4	*Fusarium graminearum* (FG2)	46	45	39	38	43
5	*Trichoderma virens*	45	45	30	29	23
6	*Fusarium solani*	17	18	26	25	25
7	*Trichoderma harzianum* (TH1)	4	16	24.5	44	45
8	*Trichoderma harzianum* (TH2)	31	33	32.5	31	37
9	*Rhizoctonia solani* (RS1)	34	74	64	33	33
10	*Rhizoctonia solani* (RS2)	58	53	38	36	18
11	*Rhizoctonia solani* (RS3)	44	47	33	24	24
12	*Rhizoctonia solani* (RS4)	73	60	42	40	35
13	*Rhizoctonia solani* (RS5)	53	51	45	38	34
14	*Aureobasidium pullulans*	47	35	27	26	28
15	*Sclerotinia sclerotiorum* (SS1)	31	38	33	32	30
16	*Sclerotinia sclerotiorum* (SS2)	33	37	36	32	20
17	*Verticillium*	39	33	27	28	27
18	*Alternaria* sp. (A1)	47	48	48	45.5	43
19	*Alternaria* sp. (A2)	12	14	19	46.5	42
20	*Chaetomium. globosum*	22	34	33	33	38
21	*Penicillium funiculosum*	23	25	28	35	37
22	*Botrytis cinerea* (BC1)	12	18	28	47	48
23	*Botrytis cinerea* (BC2)	39	43	35	38	38
24	*Alternaria raphani*	18	25	25	28	36

**Table 2 molecules-27-03336-t002:** Percentage of cyanide degradation by *T. harzianum* in different cultures with four cyanide concentrations.

Cyanide Concentration	3-Day Culture	5-Day Culture	7-Day Culture
15 mM	37%	40%	75%
20 mM	30%	38%	40%
25 mM	10%	10%	13%
30 mM	10%	9%	8%

**Table 3 molecules-27-03336-t003:** The cyanide hydratase gene and protein alignment in BLAST and its identity with other homologous sequences.

Organism	Accession Number of Cht Gene	Identity	Query Cover	Accession Number of Cht Protein	Identity	Query Cover
*Trichderma simmonsii*	CP075866.1	94.01%	99%	QYS99975.1	96.97%	100%
*Trichderma harzianum* CBS 226.95	XM_024916132.1	99.28%	81%	XP_024773123.1	98.62%	100%
*Trichderma virens* Gv29-8	XM_0140973141.1	91.37%	56%	XP_013952789.1	95.82%	98%
*Trichoderma reesei* QM6a	XM_006966895.1	89.46%	48%	XP_006966957.1	92.22%	98%
*Trichoderma citrinoviride*	XM_024891496.1	88.15%	38%	XP_024746524.1	91.94%	98%
*Isaria fumosorosea* ARSEF 2679	XM_018851715.1	79.01%	25%	XP_018701201.1	79.89%	98%

**Table 4 molecules-27-03336-t004:** Comparison of the rate of cyanide biodegradation through different microorganisms.

Microorganism	Cyanide Degradation	Description	References
*A. niger* N10	80%	Cyanide hydratase expression in prokaryotic host and its purification and biodegradation in 25 mM cyanide concentration.	[20]
*F. oxysporum*	96%	Immobilized fungal strain in reactor and biodegradation of 1 to 7 mM cyanide concentration.	[30]
*F. solani*	90%	Decomposition of 1 mM cyanide using cyanide hydratase and amidase in ammonia and its usage for greater growth.	[31]
*Scenedesmus obliquus*	92%	Biodegradation of 3 mM cyanide concentration.	[32]
*Agrobacterium tumefaciens* SUTS 1	87.5%	Biodegradation of 1, 2, and 6 mM cyanide concentration,	[16]
*Rhodococcus* UKMP-5M	94%	Biodegradation of 12 mM cyanide concentration using immobilized cells.	[33]
*Basidiomycota*	100%	Biodegradation of 4 mM cyanide concentration using 3 g fungal biomass.	[34]
*Fusarium* sp.	100%	Cyanide hydratase enzyme production and its usage for biodegradation of 15 mM cyanide concentration.	[35]
*T. koningii*	100%	Making mutations in cyanide hydratase gene to improve its activity.	[36]
*T. harzianum*	75%	15 mM cyanide concentration in a 7-day culture.	This study

**Table 5 molecules-27-03336-t005:** Fungal strains.

No.	Fungus	No.	Fungus
1	*Fusarium oxysporum* (FO1)	13	*Rhizoctonia solani* (RS5)
2	*Fusarium oxysporum* (FO2)	14	*Arebasidia pullulans*
3	*Fusarium graminearum* (FG1)	15	*Schlerotinia sclerotiorum* (SS1)
4	*Fusarium graminearum* (FG2)	16	*Schlerotinia sclerotiorum* (SS2)
5	*Trichoderma virens*	17	*Verticillium*
6	*Fusarium solani*	18	*Alternaria* sp. (A1)
7	*Trichoderma harzianum* (TH1)	19	*Alternaria* sp. (A2)
8	*Trichoderma harzianum* (TH2)	20	*Cheatominum globosum*
9	*Rhizoctonia solani* (RS1)	21	*Penicillium funiculosum*
10	*Rhizoctonia solani* (RS2)	22	*Botrytis cinerea* (BC1)
11	*Rhizoctonia solani* (RS3)	23	*Botrytis cinerea* (BC2)
12	*Rhizoctonia solani* (RS4)	24	*Aphanomyces raphani*

**Table 6 molecules-27-03336-t006:** Sequence and length of primers.

Primers	Sequence 5′ to 3′	Length of Primers
TCynF	5′-ATGGTTGCCACTATTCGTC-3′	19
TCynR1	5′-TTATTCCTTCTTCTTCTCCTC-3′	21
TCynR2	5′-CTACTCCTCAGGTGCAGG-3′	18

## Data Availability

Not applicable.

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
