# Peer review of "Cyanide Biodegradation by Trichoderma harzianum and Cyanide Hydratase Network Analysis"

_molecules, 2022, doi:10.3390/molecules27103336_

Round 1

Reviewer 1 Report

The present manuscript entitled “Cyanide biodegradation by Trichoderma harzianum and cyanide hydratase network analysis” reports some interesting results about cyanide biodegradation and cyanide hydratase network. In view of some obvious lack found in the paper, it should be carefully improved before publication. Below are some comments:

  1. Abstract should be written more precisely and explain novelty of this work.
  2. Some descriptions in the introduction are redundant, please use concise words to express your point of view. This part should focus on the research progress related to the topic and emphasize the innovation of this research. However, the novelty and significance of the topic were not highlighted, please modify the introduction more clearly.
  3. Line 27-80: It is welcome to re-construct this paragraph based on a logic order.  This paragraph islonger than two pages. Long paragraph leads to the lack of focus on the central content.
  4. Table 1: please use full name for the genus name for the first time. “s”, should notwritten in italics. 
  5. Please check that abbreviations/acronyms are defined the first time they appear in each of three sections: the abstract; the main text; the first figure or table.
  6. Figure 1: some parts are missing.
  7. Figure 1: 2: please use the same format throughout the manuscript.
  8. 2: T. harzianum, use full name for the genus name.
  9. Line 123: in the 7-day,(Table 2). should be in the 7-day (Table 2).
  10. Table 2: should keep one significant digit for the data.
  11. 5: unclear. Please re-draw the figure.
  12. Discussion: The poor discussion of the results. Author just shows the great amount of results that they have achieved, but they did not use them to develop an interesting discussion which could supplement to earlier studieson cyanide biodegradation.
  13. Line 201-231: Rephrase thelong paragraph. It is welcome that the authors discuss with the results in a logic order. 
  14. Table 6: sequence and length of primers. Can be put as Supplementary Materials.
  15. Please check the name of the microbes should be used in italics in all the reference
  16. Authors can add and revise the Conclusions section for the better understanding of the topic and its future research.
  17. There are severalmistakes appeared throughout the review manuscript including grammatical errors that need to be fixed.

Author Response

Reviewer 1

Abstract should be written more precisely and explain novelty of this work

 The abstract was modified as suggested by the reviewer.

Some descriptions in the introduction are redundant, please use concise words to express your point of view. This part should focus on the research progress related to the topic and emphasize the innovation of this research. However, the novelty and significance of the topic were not highlighted, please modify the introduction more clearly.

The introduction has been rearranged and some information added to make sense of the point that is being expressed. The editions are highlighted in red.

Line 27-80: It is welcome to re-construct this paragraph based on a logic order. This paragraph is longer than two pages. Long paragraph leads to the lack of focus on the central content.

This has been re-constructed such that the meaning of the sentences are expressed clearly.

Table 1: please use full name for the genus name for the first time. “s”, should not written in italics.

The table has been rectified and realigned as suggested.

Please check that abbreviations/acronyms are defined the first time they appear in each of three sections: the abstract; the main text; the first figure or table

The abbreviations have been checked and corrected throughout the manuscript as suggested.

Figure 1: some parts are missing.

This comment is unclear. Can the reviewer please clarify this comment.

Figure 2: please use the same format throughout the manuscript.

The format of figure 1 has been amended to align with that of figure 2.

T. harzianum, use full name for the genus name

This has been rectified as first appearance but used as a shortened version thereafter.

Line 123: in the 7-day,(Table 2). should be in the 7-day (Table 2).

This has been corrected

Table 2: should keep one significant digit for the data.

The authors would like to plead to the reviewer that this interpretation of the data be kept as is as it suits the style that the authors have chosen for the representation of their data.

Figure `5: unclear. Please re-draw the figure.

Figure 5 is the phylogenetic tree. The authors feel that the figure is clear enough since a software that constructs it was used.

Discussion: The poor discussion of the results. Author just shows the great amount of results that they have achieved, but they did not use them to develop an interesting discussion which could supplement to earlier studies on cyanide biodegradation.

The discussion has been improved such that previous studies are correlated with the data that was observed in this study.

Line 201-231: Rephrase the long paragraph. It is welcome that the authors discuss with the results in a logic order.

The discussion has been re-worked and realigned with the inclusion of sentences such that the message being communicated is clearer.

Table 6: sequence and length of primers. Can be put as Supplementary Materials.

Normally, in papers that represent molecular based techniques, it is important to include the primers on actual manuscript

Please check the name of the microbes should be used in italics in all the reference

This has been corrected and the names are written in italics.

Authors can add and revise the Conclusions section for the better understanding of the topic and its future research.

The whole conclusion section has been revised and future research was also included

There are several mistakes appeared throughout the review manuscript including grammatical errors that need to be fixed.

These have been detected by the authors and fixed accordingly

Reviewer 2 Report

The authors based their work on the very interesting assumption that fungi may be effective in cyanide bioremediation, where they defined the gene responsible for the cyanide detoxification. Using adequate genetic and bioinformatics tools, a network of connections between the genes of cyanide hydratase and some related genes was constructed. They also demonstrated a key role of cyanide hydratase in the degradation of cyanides.It is extremely important that the highest tolerance to cyanide and the ability to use it as a source of C and N biodegradation efficiency was achieved for the Trichoderma harzianum (T1) strain, representing a species with great biostimulation and biocontrol potential.

It should be emphasized that strains of F. oxysporum achieving high (similar to T. harzianum (T1) biomass in the presence of cyanide can also be non-pathogenic and biocontrol strains and their use in the biodegradation of cyanides is also possible.

Do the authors have data on the pathogenicity of F. oxysporum to plants? This should be mentioned.

Why were strains belonging to rhizosphere fungi species and having various interactions with plants selected for the study? Is it planned to combine phytoremediation and bioaugmentation methods? T. harzianum could certainly be used in bioaugmentation.

The possibility of using plants (what species?) Colonized by T. harzianum (T1) should be mentioned.

The work also provides information on the enzyme complex responsible for the subsequent stages of degradation of cyanide hydratase, dipeptidase, carbon-nitrogen hydrolase-like protein, and ATP adenylotransferase, where the enzymes do not need any cofactors, the triple non-peptide bonds in cyanide compounds are cleaved directly and the products have low toxicity and can be degraded further.

What was the composition of the PDB medium that was tested for cyanide tolerance. The composition must be specified in the Materials and methods chapter.Did it really not contain C and N source? On what basis do the authors argue that cyanide was used as a source of C and N?When discussing the results, authors should not repeat the information contained in the materials and methods chapter.Line 70 saprotrophic instead of saprophyticLine 102 phytopathogen instead of pathogenTable 1The table title is poorly formulated. Should read: Biomass of 24 fungal isolates cultured in four different cyanide-containing media (2, 5, 10, and 15 mM cyanide) achieved on day 7 of incubationabbreviation sp. should not be written in italicsFig. 1.What the authors mean by the word "proportion" is quite unclear. Does this need to be clarified both in the text and in the title of Fig.1Table 4. If the authors wanted to compare, in a rather unusual way, to compare the effectiveness of bioremediation obtained in their own research with the effectiveness obtained by other researchers in the discussion using the table, then the conversion to the same units (mM) of cyanide concentration should be used in the description columnLine 265 - inoculum size determination (1 mm2 of harvested spores and mycelia)is very imprecise - please enter them in a number of spores determined in the hemocytometer and the number of CFUs determining the cultureLine 272 "selected based on its non-pathogenicity to plants" - how was pathogenicity determined?Line 278-280 - please describe in detail the method for the determination of the cyanide concentrationLine 339 capability not capbility

Author Response

Reviewer 2

It should be emphasized that strains of F. oxysporum achieving high (similar to T. harzianum (T1) biomass in the presence of cyanide can also be non-pathogenic and biocontrol strains and their use in the biodegradation of cyanides is also possible.

This has been covered in the conclusion section

Do the authors have data on the pathogenicity of F. oxysporum to plants? This should be mentioned

The authors do not have data but the authors consulted literature to determine the pathogenicity of F. oxysporum (see 10.1111/j.1364-3703.2009.00538.x)

Why were strains belonging to rhizosphere fungi species and having various interactions with plants selected for the study? Is it planned to combine phytoremediation and bioaugmentation methods? T. harzianum could certainly be used in bioaugmentation.

The possibility of using plants (what species?) Colonized by T. harzianum (T1) should be mentioned.

The plan is to utilise both phytoremediation and bioaugmentation, but phytoremediation was not the focus of the current study. A variety of plants including natural and transgenic are associated with rhizosphere micro-organisms such as fungi and others. However, some fungal organisms can be pathogenic to plants, and this will defeat the objective of utilisation phytoremediation and bioaugmentation for the removal of cyanide species. Hence this focused on a non-pathogenic fungus to remediate cyanide. plants. In this condition, both of phytoremediation and bioaugmentation can be used and they can have synergistic effect.

What was the composition of the PDB medium that was tested for cyanide tolerance. The composition must be specified in the Materials and methods chapter.

The composition of PDB has been included under section 4.2 as follows “The PDB contained 4 g/L potato powder and 20 g/L of dextrose.”

Did it really not contain C and N source? On what basis do the authors argue that cyanide was used as a source of C and N?

The writing has been amended to represent that cyanide is used as a nitrogen source since the composition of the PDB is carbon-based (starch and dextrose).

When discussing the results, authors should not repeat the information contained in the materials and methods chapter.

This has been corrected. The discussion has been rearranged and corrected.

Line 70 saprotrophic instead of saprophytic

This has been corrected as suggested by the reviewer.

Line 102 phytopathogen instead of pathogen

This has been corrected as suggested by the reviewer.

Table 1: The table title is poorly formulated. Should read: Biomass of 24 fungal isolates cultured in four different cyanide-containing media (2, 5, 10, and 15 mM cyanide) achieved on day 7 of incubation abbreviation sp. should not be written in italics

The table has been renamed and corrected as suggested by the reviewer. Thank you for the suggestion.

Table 4. If the authors wanted to compare, in a rather unusual way, to compare the effectiveness of bioremediation obtained in their own research with the effectiveness obtained by other researchers in the discussion using the table, then the conversion to the same units (mM) of cyanide concentration should be used in the description column

The units have been changed to the same units (mM).

Line 265 - inoculum size determination (1 mm2 of harvested spores and mycelia) is very imprecise - please enter them in a number of spores determined in the hemocytometer and the number of CFUs determining the culture

The inoculum size determination has been corrected and more information added for clarity as follows “All the organisms were cultured in Potato Dextrose Agar in petri dishes for 14 days in at room temperature. After the 14-day period, the plates were flooded with sterile distilled water and the mycelia was scraped from the surface of the agar using a sterile inoculation loop. The mycelia was separated from the liquid fraction (contained the spores) by filtration using Whatman no.1 filter paper and were inoculated in were inoculated in cyanide containing Potato Dextrose Broth (PDB) at a concentration of 3 x 106 spores/L. The spore concentration was calculated using a Hemacytometer under an Optical Microscope (Olympus, South Africa).”

Line 272 "selected based on its non-pathogenicity to plants" - how was pathogenicity determined?

A sentence has been included as follows “The non-pathogenicity of the organisms was determined by consulting literature.”

Line 278-280 - please describe in detail the method for the determination of the cyanide concentration

Cyanide was determined according to the procedure that was set out by Fisher and Brown (1952) as referenced in the manuscript. The detailed description of the method is evident in that publication and the authors decided not to repeat the information that is already described in that publication.

Line 339 capability not capbility

This error has been fixed

Round 2

Reviewer 1 Report

The authors have considered all comments raised by the reviewers and revised the manuscript accordingly based on these comments. The revision is fine and can be accepted for publication in current form.